# A robust transcriptional program in newts undergoing multiple events of lens regeneration throughout their lifespan

Konstantinos Sousounis[1], Feng Qi[2], Manisha C Yadav[3], José Luis Millán[3], Fubito Toyama[4], Chikafumi Chiba[5], Yukiko Eguchi[6†], Goro Eguchi[6*], Panagiotis A Tsonis[1,3*]

[1]Department of Biology, University of Dayton, Dayton, United States; [2]Sanford Burnham Prebys Medical Discovery Institute at Lake Nona, Orlando, United States; [3]Sanford Children's Health Research Center, Sanford-Burnham-Prebys Medical Discovery Institute, La Jolla, United States; [4]Graduate School of Engineering, Utsunomiya University, Utsunomiya, Japan; [5]Faculty of Life and Environmental Sciences, Tsukuba University, Tsukuba, Japan; [6]National Institute for Basic Biology, National Institutes for Natural Sciences, Okazaki, Japan

**Abstract** Newts have the ability to repeatedly regenerate their lens even during ageing. However, it is unclear whether this regeneration reflects an undisturbed genetic activity. To answer this question, we compared the transcriptomes of lenses, irises and tails from aged newts that had undergone lens regeneration 19 times with the equivalent tissues from young newts that had never experienced lens regeneration. Our analysis indicates that repeatedly regenerated lenses showed a robust transcriptional program comparable to young never-regenerated lenses. In contrast, the tail, which was never regenerated, showed gene expression signatures of ageing. Our analysis strongly suggests that, with respect to gene expression, the regenerated lenses have not deviated from a robust transcriptional program even after multiple events of regeneration throughout the life of the newt. In addition, our study provides a new paradigm in biology, and establishes the newt as a key model for the study of regeneration in relation to ageing.

*For correspondence: goro-eguchi@fj8.so-net.ne.jp (GE); ptsonis1@udayton.edu (PAT)

†Deceased

**Competing interests:** The authors declare that no competing interests exist.

## Introduction

Newts are among the few vertebrates that possess the remarkable ability to regenerate tissues, organs, and body parts, including limbs, tails and eye tissue (*Sanchez Alvarado and Tsonis, 2006*). Importantly, newts appear to regenerate using cells recruited locally from the site of the insult. For example, during limb regeneration, the cells at the site of amputation, such as muscle and bone cells, dedifferentiate and then redifferentiate to reconstruct the lost part (*Tsonis, 1996*; *Sandoval-Guzmán et al., 2014*). For this process to occur, cells from the original organ must remain to provide a source for regeneration; if the entire limb is removed, no regeneration occurs. However, regeneration of the lens is different in two key ways, providing additional experimental benefits. First, regeneration is possible following complete removal of the lens, and thus whole-organ regeneration occurs. Second, the lens is regenerated from a different tissue, that is, the pigment epithelial cells (PECs) of the dorsal iris, via transdifferentiation rather than from the remaining lens tissue (*Henry and Tsonis, 2010*; *Barbosa-Sabanero et al., 2012*). Because of these unparalleled regenerative traits, newts may provide answers that regenerative medicine is presently seeking (*Baddour et al., 2012*). A fundamental question is whether newt regenerative ability

**eLife digest** Newts are unusual animals because they are able to regenerate injured or lost body parts. To regenerate the lens in an eye, certain cells in the iris need to change into lens cells. In 2011, a group of researchers reported the results of a 16-year long study of lens regeneration in Japanese newts. This study found that lenses from old newts that have undergone lens regeneration many times are structurally identical to those of young individuals that still have their original lenses. Also, many genes required to make lens proteins were expressed at similar levels in the lenses of the old and young newts. Therefore, even old newts retain the ability to fully regenerate their lenses.

However, it is possible that the lenses in the old newts might show more subtle signs of ageing in the form of differences in the expression of other genes. Here, Sousounis et al. – including some of the researchers from the 2011 work – used an approach called transcriptomics to examine the patterns of gene expression in this group of newts in more detail.

Sousounis et al. collected cells from the lenses, irises and tails of both the old and young newts. The experiments show that the patterns of gene expression in the regenerated lenses closely resemble the patterns seen in the lenses of the young newts. In contrast, the tail cells of the old and young newts display different gene expression patterns, with those from the older newts displaying hallmarks of ageing that are absent in the younger newts. The iris cells from the old newts show a mixed gene expression profile with features characteristic of both young and aged tissue. Sousounis et al.'s findings highlight the value of using newts as models to study the links between regeneration and ageing

declines with age or repeated insult. To answer this question, we undertook a long-term study of lens regeneration.

Using Japanese newts (*Cynops pyrrhogaster*), lens regeneration was followed for 16 years. During this period, lenses were removed from the same animals 18 times. Previously, it was shown that the 17 and 18 times regenerated lenses, which were obtained from the second-to-last and last collections, respectively, were virtually identical to the intact lenses removed from full-grown, 14-year-old newts produced from fertilized eggs that had never undergone a lentectomy or lens regeneration. Throughout this 16-year period, the rate and stage of regeneration was carefully evaluated, and no significant delay in the lens regeneration process was observed in any of the 18 repetitions (*Eguchi et al., 2011*). At the gross anatomical level, the experimental and control lenses were of the same size and transparency. The lens fiber organization appeared normal, with the nucleus containing primary fibers and the cortex containing secondary fibers. Most importantly, the gene expression patterns of the experimental and control lenses were very similar. The genes examined included crystallins and transcription factors that regulate crystallin expression, such as Pax-6, Sox2, MafB, Sox1, Prox-1, and Delta, all of which participate in lens development and lens fiber differentiation and are thus involved in normal lens homeostasis. The study also established that the age of the animal does not affect its regenerative capacity (see also *Sousounis et al., 2014*). The newts were estimated to be at least 14 years old at the onset of the project and thus would have been at least 30 years old at the end of the study. Because the reported lifespan of the Japanese newt is 25 years (*Goin et al., 1978*), this group truly represents an old population. These results raise the question as to whether the repeatedly regenerated lens of a 30-year-old newt retains the biological signature of a 14-year-old's lens. Especially this is of interest if one considers the relation of regeneration and ageing. To investigate this possibility, we undertook a transcriptomic analysis of lenses that had been regenerated 19 times along with appropriate controls.

## Results

### Samples

The Japanese newt *C. pyrrhogaster* was used in this study. The experimental and control groups of newts were as follows. The experimental group (referred to as #19 throughout) comprised of 32-

year-old newts whose lenses had been removed 19 times. These lenses were regenerated 19 times and removed 18 years after the start of the project. The control group (referred to as #0) consisted of 14-year-old newts that had their original lenses (i.e., non-regenerated lenses) (*Figure 1*). The tissue collected from the experimental group consisted of #19 lenses, #19 dorsal irises, and #19 tails (n=5 for each tissue type). The dorsal iris was sampled because this tissue gives rise to the regenerated lens, which implies that the dorsal iris had also been regenerated/replenished 19 times. The tails were included as an aged tissue that had never been regenerated. The corresponding tissues were also sampled from #0 newts. In total, 30 samples were prepared for RNA sequencing and transcriptomic analysis.

## Sequencing and annotation

We generated nearly 4.5 billion reads, with approximately 150 million reads per sample. The reads were of high quality (>97% passed TAILING:30 criteria using Trimmomatic [*Bolger et al., 2014*]) and included very few duplicates (approximately 2%, as assessed using FastUniq [*Xu et al., 2012*]). Trinity was used for de novo assembly of the reference transcriptome (see Materials and methods), which was composed of 4.3 million contigs and isoforms (referred to as transcripts or genes throughout). We used NCBI BLASTx to annotate 133,503 (73,233 contigs) of the transcripts against the human reference proteome obtained from UniProt (e-value<1E-10). Remarkably, 58,331 of these transcripts were related to human transposons (43.7%). In total, we obtained 15,077 non-redundant annotations representing nearly 75% of all human genes.

## Tissue-specific enriched gene expression

Reads were used to compute the relative abundance of transcripts in each sample. Transcripts that showed the most significant variability between samples are shown as a heat map (*Figure 2A*). To identify highly expressed genes in the three different tissues we focused only on the annotated transcripts and considered the ones with >1000 fragment per kilobase per millions of reads (FPKM). In other words, which were the genes with the highest expression in each tissue irrespective of treatment (young or old) (*Figure 2B*, *Supplementary file 1*). As expected alpha-, beta-, and gamma-crystallin genes (CRY) were found to be the highest expressed genes in lens samples. Crystallins are known to be the major structural protein of the lens (*Masters et al., 1977*). The same dataset also contained the lens fiber major intrinsic protein MIP, and phakinin (BFSP2), genes highly expressed in lenses (*Figure 2B*; orange) (*Broekhuyse and Kuhlmann, 1974*; *Maisel and Perry, 1972*). Ornithine decarboxylase antizyme 1 (OAZ1), hemoglobin subunit alpha (HBA1), and cell division control protein 42 homolog (CDC42) were the highest expressed genes in the iris samples (*Figure 2B*; red). Expectedly, keratin (KRT) and ribosomal protein genes were the ones with the highest expression in tails (*Figure 2B*; yellow). Keratin proteins are known to be expressed in the skin. Six genes, five coding for ribosomal proteins and one for the ferritin heavy chain, were found to be the most expressed in all tissue types (*Figure 2B*; purple). In a different analysis, we identified genes exclusively or preferentially expressed in a particular tissue, when compared with the others. We sorted genes that were adequately expressed in a given tissue (FPKM>100) and were 100-fold more expressed in one

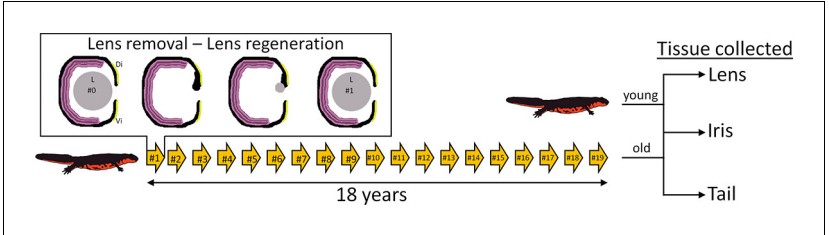

**Figure 1.** Experimental overview. Arrows depict the number of repeated lentectomies performed over a period of 18 years. Panel shows the process of lens regeneration that occurred after each lens removal highlighted as a single arrow. At the end of the experiment, lens, iris, and tail tissues were collected from both old newts that had regenerated their lenses 19 times and young newts that had never experienced lentectomy. Di: Dorsal iris; Vi: Ventral iris; L: Lens.

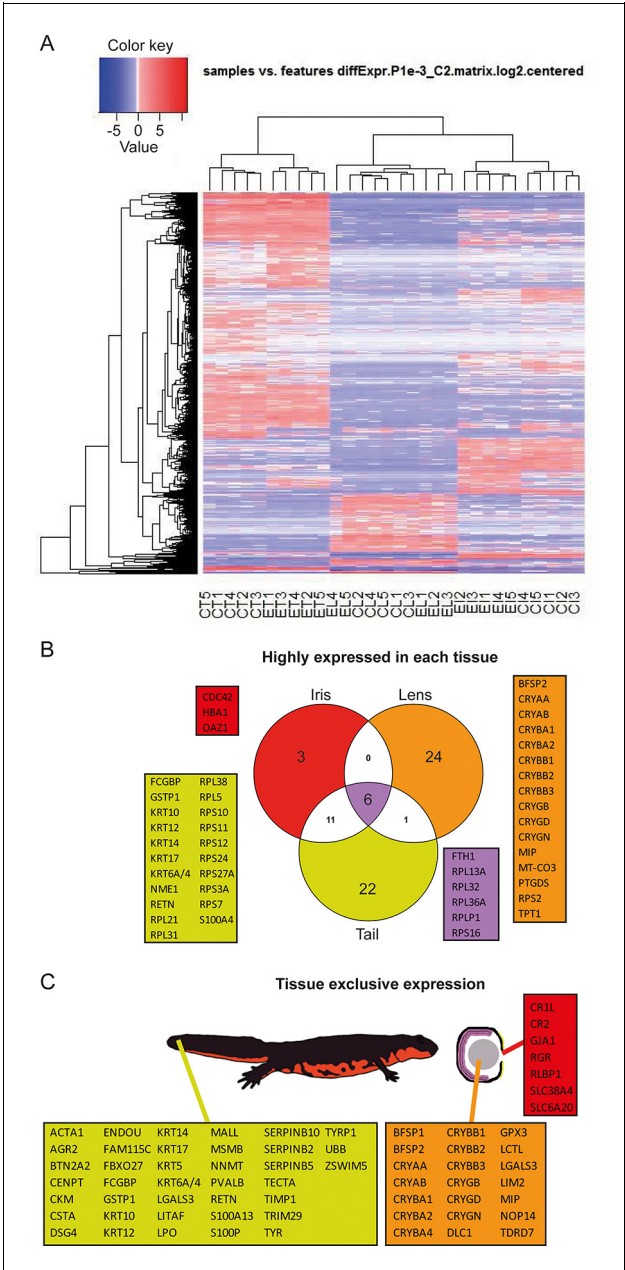

**Figure 2.** Gene expression among tissue samples. (**A**) Heat map constructed from the expression profiles of the 30 sequenced samples. CT, CL, CI: #0 (control) tail, lens, and iris, respectively. ET, EL, EI: #19 (experimental) tail, lens, and iris, respectively. Genes selected based on the following parameters: p-value<=0.001 and log$_2$(FC)>=2. Note the nearly uniform pattern between the #0 and #19 lens samples which indicates no differences between non-regenerated young lenses and repeatedly regenerated lenses from aged newts. (**B**) Highly expressed genes in each tissue type irrespective of age. Red, orange, and yellow colors denote genes from iris, lens, and tail samples, respectively. Comparisons are visualized using a venn graph while non-redundant annotations are highlighted using boxes including the corresponding gene names. Purple color is used for highly expressed genes in all three samples. (**C**) Genes that are preferentially expressed in a given tissue versus the others are denoted using the same color code as in B. The different tissues are indicated using a cartoon newt and an enlarged cross-sectioned eye.

versus the other tissues (*Figure 2C*, *Supplementary file 1*). Genes in the lens dataset included crystallins, lens fiber membrane intrinsic protein LIM2, filensin (BFSP1), and tudor domain-containing protein 7 (TDRD7), among others (*Figure 2C*; orange). LIM2, BFSP1, and TDRD7 are known to be expressed in lenses (*Church and Wang, 1993*; *Hess et al., 1998*; *Lachke et al., 2011*). Iris preferentially expressed retinal pigment epithelial (RPE)-retinal G protein-coupled receptor (RGR), a protein found in pigmented cells of the retina (*Figure 2C*; red) (*Tao et al., 1998*), while tail samples expressed keratins, creatine kinase M-type (CKM), resistin (RETN), and alpha skeletal muscle actin (ACTA1) (*Figure 2C*; yellow), proteins found in muscle, skin, and adipose tissue (*Way et al., 2001*; *Nowak et al., 1999*). Many of the genes identified by these two methods are known to be enriched in the same or equivalent (e.g., tail to be a composition of muscle, fat, skin, and spinal cord) tissues in other organisms including humans. These genes are also known to be involved with disease states including lens cataracts or ageing.

## Analysis of differential gene expression between #0 and #19 samples

Differential gene expression analysis between #0 and #19 equivalent tissues was performed using edgeR (*Robinson et al., 2010*). This analysis provided us with genes that their abundance changed during ageing and repetitive lens regeneration. No genes were found to differ significantly in their expression between the #19 and #0 lens samples (false discovery rate [FDR] <0.05 and fold change [FC] >2; *Supplementary file 2*). In the iris samples, we found 311 (54 of these annotated) genes with FDR<0.05 and FC>2 (*Figure 3A* and *Supplementary file 3*). Even greater differences in gene expression were observed for the tail samples. We found 4204 (780 of these annotated) genes with FDR<0.05 and FC >2 (*Figure 3B* and *Supplementary file 4*). In our experimental design, tail samples were collected in order to provide a tissue that was never amputated or regenerated from the same animals where repetitive lentectomy was performed during the last 19 years. Gene expression comparisons between young #0 tails and old #19 tails were conducted to prove that amphibian gene expression signatures change over time as tissues age. To begin with, we studied the roles of the differentially regulated genes in the tail samples by assigning Gene Ontology (GO) terms based on their biological processes, molecular functions, and sub-cellular localization (*Supplementary file 5*). Enrichment analysis revealed that GO terms related to translation, electron transport chain, oxidation reduction, and mitochondrion were enriched in the group of down-regulated genes in the #19 tail samples (*Figure 3C*; green bars; FDR<0.05, *Supplementary file 5*). As will be discussed later, iris samples also showed enrichment of electron transport chain genes in the down-regulation dataset. Down-regulation of electron transport chain-associated genes is a well-established signature of ageing in many vertebrate animal models and flies (*López-Otín et al., 2013*; *Zahn et al., 2006*). To further illustrate this, we data-mined genes with GO terms related to ageing and/or senescence, which were differentially regulated between #0 and #19 tail samples and found 16 genes (*Figure 3D*). These data suggest that the transcriptomic profile of newt tails and irises changed over time and showed signs of ageing. Since we observed changes in the abundance of several genes in tail and iris samples, we next asked whether these changes were reflected in the transcriptomic complexity of these tissues. By sorting genes based on their relative abundance we plotted the percent contribution of each gene in a cumulative way (*Figure 3E*, *Supplementary file 1*). This method identifies how many genes are sharing the total transcriptomic output of each sample; for example, if 50% of the transcriptomic output is shared by 100 genes, the underlying profile is relatively simple and the line on the plot will appear flat (orange; *Figure 3E*). On the other hand, if 1000 genes are sharing 50% of the transcriptomic output, the profile of the tissue is more complex having a steeper line (red; *Figure 3E*) (*Mele et al., 2015*). This analysis reveals that iris is the most complex tissue followed by the tail and the lens (*Figure 3E*). Interestingly, #19 iris and tail samples showed slightly increased complexity versus the respective #0 samples. #19 lens samples showed the lowest increase corroborating our previous data indicating no significant changes between #0 and #19 lens samples maintaining a stable gene expression profile. Using sample correlation matrix plot we further validated our gene expression data (*Figure 4A*, *Supplementary file 6*).When the genes from the #19 lenses (EL1–EL5) were compared with those from the #0 lenses (CL1–CL5), a nearly uniform pattern was obtained across all 10 samples. This pattern similarity was also evidenced by the lack of segregation of the experimental and control lens samples via cladograms. These results indicate that these samples were highly correlated for their overall gene expression pattern. However, a different pattern emerged when comparing the irises and tails between the #19 and #0 groups. Areas on the

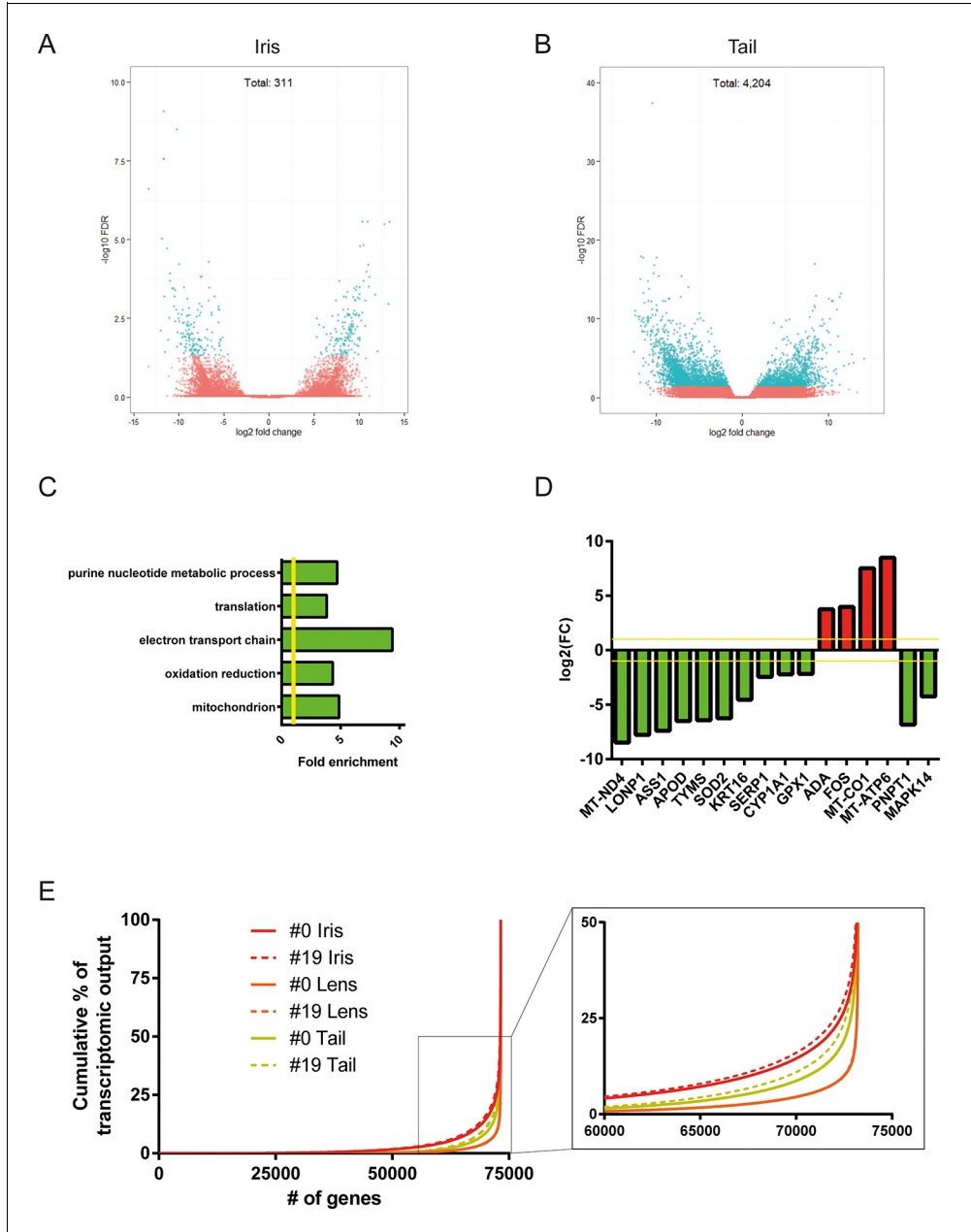

**Figure 3.** Differential gene expression between #19 and #0 tissues. (A) Volcano plot for the #19 versus #0 iris samples. (B) Volcano plot for the #19 versus #0 tail samples. Differentially expressed genes (false discovery rate [FDR]<0.05 and fold change [FC]>2) are depicted in cyan. Tail samples, which never experienced regeneration, showed the most differentially expressed genes. Iris samples, which as the source of lens regeneration have experienced some degree of regeneration/replenishment, showed a reduced number of differentially expressed genes and an intermediate ageing profile. (C) Selected enriched (FDR <0.05) Gene Ontology (GO) terms in tail samples plotted based on their fold enrichment. Green-colored bars mark gene groups that are down-regulated in the #19 samples. Yellow line marks fold enrichment of 1. Electron transport chain is one of the functional group enriched in the down-regulated group, a well-documented ageing signature in other vertebrates. (D) Genes selected for their role in ageing and/or senescence and plotted based on their fold change between #19 and #0 tail samples. Green and red bars mark down-regulated or up-regulated genes in #19 samples, respectively. Yellow line marks log2(FC) of 1. These data provide additional evidence of ageing signs in our #19 tails samples. (E) Transcriptomic complexity between #0 and #19 tissues. Red, orange, and yellow denotes iris, lens, and tail samples, respectively. Solid and dotted lines represent tissues from #0 and #19 newts, respectively. In this graph, genes were sorted based on their expression and plotted based on their cumulative percent contribution to the overall transcriptomal output. Iris had the most complex transcriptome by having more genes contributing to the overall output (steeper line), followed by the tail and lens. Tissues from #19 newts showed slightly increased transcriptomic complexity versus their #0 counterparts (enlarged insert). Lens showed the least increase in complexity which further supports that lens regeneration is a robust process that can faithfully proceed throughout life.

sample correlation matrix plot showed the characteristic four-boxed color pattern indicating differences in the overall correlation between #19 and #0 samples (*Figure 4A*). Similar results were obtained by using jackknifing and random 20% sampling methods (*Supplementary files 7 and 8*). By dissecting the within tissue correlation values and comparing them, it was evident that the five biological replicates of each tissue had high correlation values among them (*Figure 4B*; solid colors

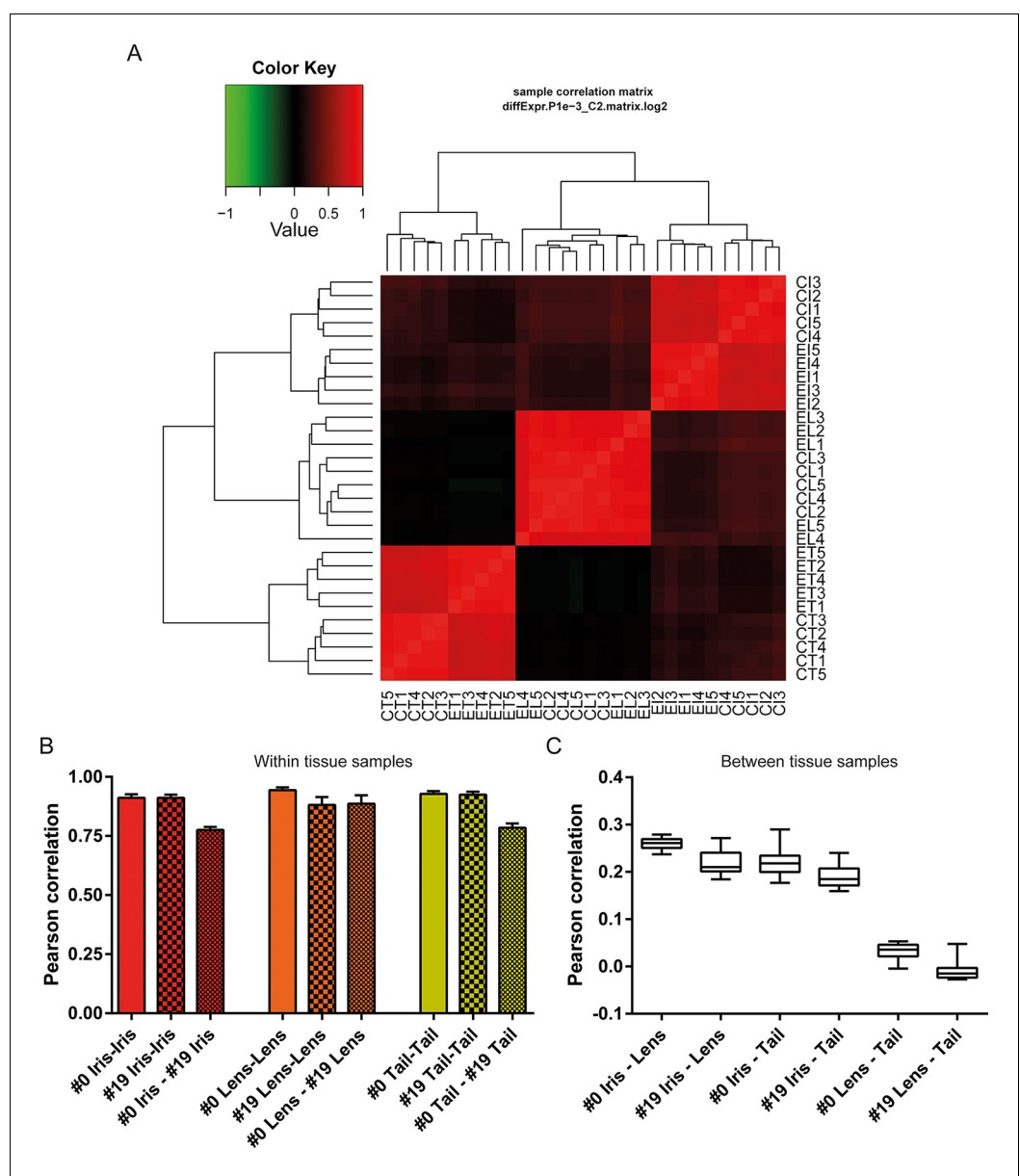

**Figure 4.** Sample correlations between the 30 samples. (**A**) Sample correlation matrix plot. Note the uniform red color between #19 (experimental lens, EL) and #0 (control lens, CL), indicating high correlation between them. Iris and tail #0 and #19 samples segregate clearly creating a characteristic four-box pattern in the two edges of the plot. #19 and #0 lens samples are so similar that the cladogram clusters them together. EL4 sample exhibits the least correlation among the #19 newt lenses. (**B**) Within tissue correlation plotted as bar graphs for better visualization. Solid colored bars (inter-#0 correlations) and big-dotted bars (inter-#19 correlations) showed very high values. Intra #0 - #19 correlation values were lower except those in lens samples. These data indicate that #0 and #19 lens samples are very similar in regards to gene expression versus equivalent comparisons in iris and tail samples. (**C**) Correlations between tissues illustrated as box plots. Iris-lens gene expression correlations were the highest followed by iris-tail and lens-tail.

and big-dotted bars). However, when comparisons are made between #0 and #19 samples (*Figure 4B*; small-dotted bars), the correlation drops with lens samples showing the least decrease (orange bars) compared to iris (red bars) and tail (yellow bars). We also performed a comparison between tissues and found that iris and lens tissues were the most related while lens and tail tissues the least related based on the genes expressed (*Figure 4C*).

Sample correlation matrix plot also revealed that sample EL4 (one from #19 old newts) did not strongly correlate with the other lens samples. The correlation values of this sample were the lowest for the within tissue comparisons performed (*Supplementary file 6*). However, the values were higher than the within tissue correlations of #0 and #19 in iris and tail samples. Nevertheless we wanted to test whether this sample showed signs of ageing. To accomplish that we isolated all crystallin-associated genes expressed in the EL4 sample and compared them to the average FPKM values of the other #19 lens samples (*Supplementary file 9*). We chose crystallin genes because crystallins are the major structural proteins of this tissue and down-regulation is often linked to disease states in humans and mice (*Sousounis and Tsonis, 2012*). Our analysis revealed that 30% of the transcripts associated with crystallin genes were deregulated in the EL4 sample. However, most of them showed higher expression in the EL4 sample than the other #19 lens samples, an expression pattern that does not match a pathological profile. For example, transcript c1474631_g1 corresponding to gamma crystallin B, a highly expressed gene in the lens, showed a more than twofold up-regulation (*Supplementary file 9*). Overall, our analysis showed that EL4 had the weakest association among the #19 lens samples and #0 young control lenses; however, differences in gene expression were not strongly associated with ageing or disease.

## Correction of age-regulated gene expression in the regenerating lens

Iris is the source of lens regeneration in newts. After lentectomy the whole lens is removed and dorsal iris PECs transdifferentiate to lens cells. By collecting iris tissue for RNA sequencing we studied how repetitive regeneration and ageing affected its transcriptomic profile. As mentioned earlier, 311 genes were found to be differentially affected when #0 and #19 iris samples were compared (*Figure 3A*, *Supplementary file 3*). Iris, as the cellular source of lens regeneration, should have reflected, at least in part, this deregulated profile to the regenerate. Surprisingly though, the lens samples did not have any significantly deregulated genes suggesting that the age-regulated profile of the iris was corrected during the regeneration process. To further highlight these differences, we compared the regulated genes by first plotting the FPKM values of #0 and #19 iris and lens samples and applying linear regression (*Figure 5A*, *Supplementary file 10*). As expected lens FPKM values were highly correlated (r = 0.9982) and differed between them on average 27% (slope; m = 0.7293) (*Figure 5A*; orange). On the other hand, iris FPKM values were not correlated and differed completely (r = 0.0070, m = 0.004, *Figure 5A*; red). These data clearly indicate that deregulation of these genes were corrected in the regenerated lens. When the function of the annotated genes was investigated, we found that electron transport chain was enriched in the group of genes that were down-regulated in the #19 iris samples (*Figure 5B*; green bars, *Supplementary file 11*). As indicated above, this is a well-established ageing signature and suggests that #19 repetitive regenerated lenses did not inherit it during the transdifferentiation process.

Overall, our results point out that repeated lens regeneration employs a robust transcriptomic program that is maintained throughout life, an attribute not found in the never-regenerated tail tissue from the same animals. In addition, the fact that #19 lenses did not show down-regulation of genes related to electron transport chain, a well-established signature of ageing revealed in #19 iris and tail samples, suggests that repeated regeneration might ameliorate age-regulated gene changes.

## Discussion

Newts have the remarkable ability to regenerate their lenses after repeated insults throughout their lifespan. In order to gain additional insights about the molecular interactions underlying this trait, we started by exploring highly and uniquely expressed genes in each of the tissues collected; iris, lens, and tail. We found that genes preferentially and highly expressed in lens or tail are known to be expressed in other vertebrates. More importantly for the lens, crystallins, phakinin, filensin, tudor domain-containing protein 7, lens fiber major intrinsic protein MIP, and lens fiber membrane intrinsic

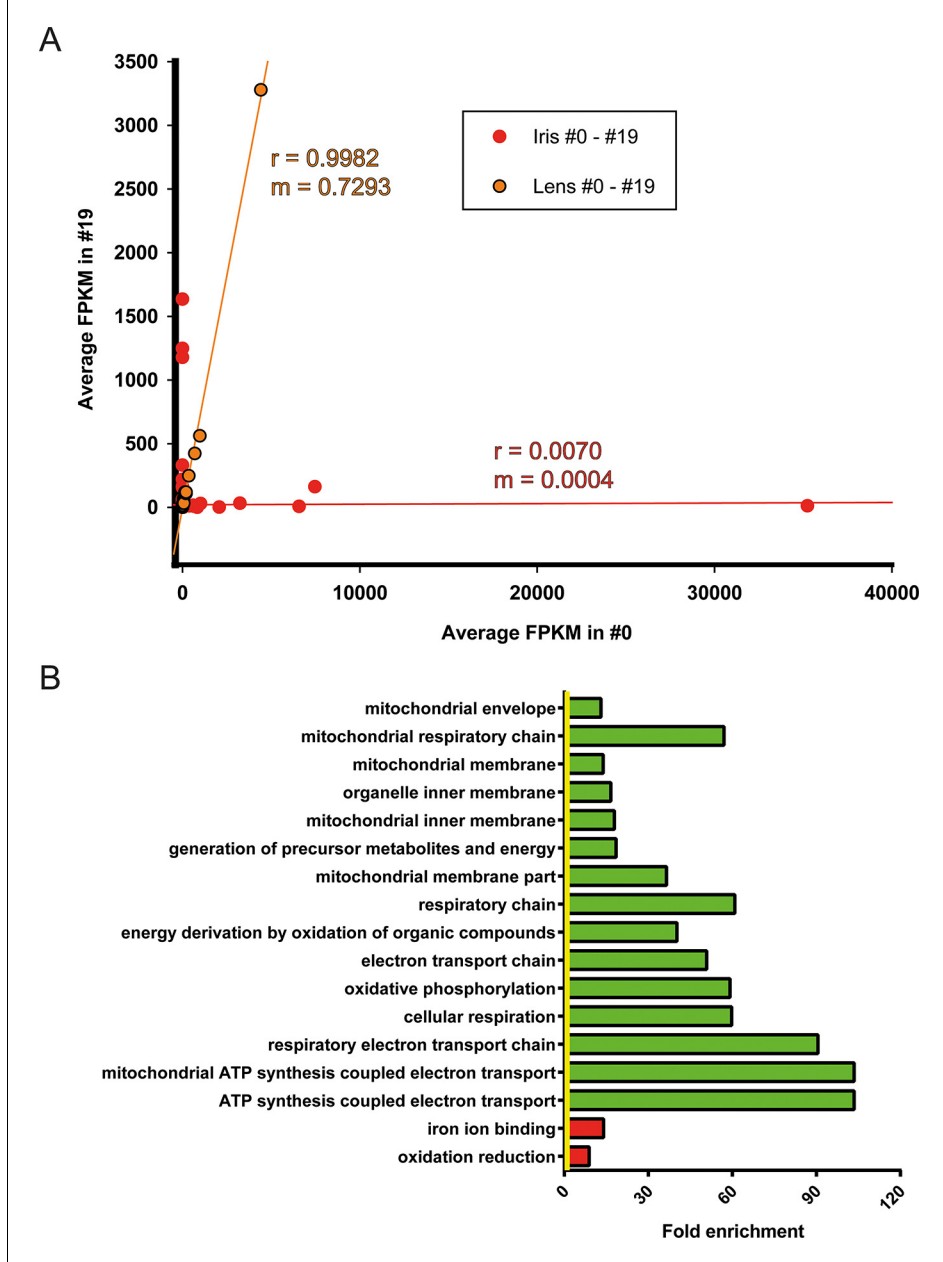

**Figure 5.** Correction of iris-regulated gene expression in the regenerated lens. (**A**) The 311 genes used for this analysis were differentially regulated in #0 versus #19 iris samples. Since lens is regenerated from the dorsal iris, the question arises whether or not these differences in gene expression are reflected in the regenerated lenses. Average fragment per kilobase per millions of reads (FPKM) values of these genes from iris and lens samples were plotted in the same graph. Red and orange colors mark the iris and lens, respectively. Linear regression analysis revealed that lens genes are more correlated (r = 0.9982) with m = 0.7293 while iris genes are not correlated (r = 0.0070) as expected. Based on the slope (**m**) values (where m = 1 is the absolute perfect when #0 and #19 values are identical), these data indicate that the #0 and #19 lens FPKM values differ approximately 27% while the equivalent iris samples are completely different. (**B**) Gene Ontology enrichment analysis of genes differentially expressed between #0 and #19 iris samples. Green and red bars denote gene groups in the down-regulated and up-regulated datasets, respectively. Yellow line marks fold enrichment of 1. Note that as with the tail samples, electron transport chain is also down-regulated in these samples, a sign of ageing.

protein LIM2 are major structural/molecular components of the lens and linked to age-related lens diseases when deregulated (*Sousounis et al., 2014*; *Lachke et al., 2011*; *Sousounis and Tsonis, 2012*; *Jakobs et al., 2000*; *Ramachandran et al., 2007*; *Berry et al., 2000*; *Pras et al., 2002* ). Thus these genes are also good markers should the newt lens age. However, when gene expression

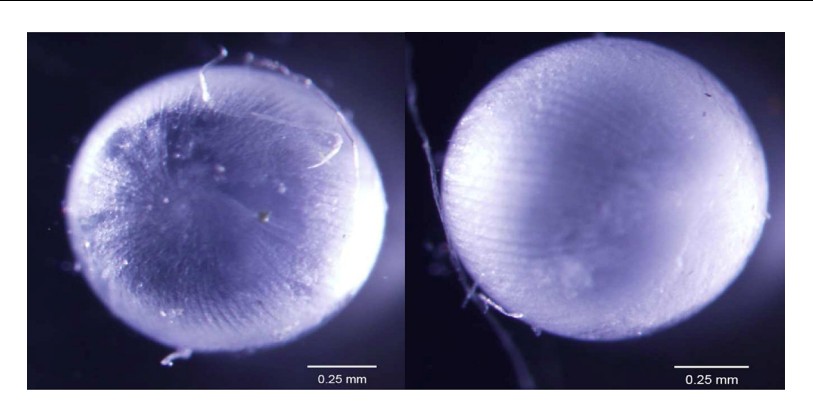

**Figure 6.** Lenses from #0 control (left) and #19 experimental (right) newts. Note that the size, fiber arrangement, and transparency of both samples are normal.

patterns of the #19 and #0 lenses were compared, no significant differences were found. In addition, compared with the non-regenerated lenses from younger animals, the #19 lenses showed no differences in size, transparency, or overall fiber structure (*Figure 6*) and during this 18-year experiment, cataracts were never observed in the regenerated lenses. This important result clearly demonstrates that repeated lentectomies and ageing have no effect on lens regeneration, development, or homeostasis. In addition to the aforementioned genetic causes of cataracts, it is well documented that lenses are severely affected by ageing with marked changes in expression of mitochondrion electron transport chain, oxidative stress and crystallin genes as well as alterations of the fiber structure and homeostasis leading to cataracts (*Su et al., 2015*; *Tsentalovich et al., 2015*; *Wei et al., 2015*; *Linetsky et al., 2014*; *Petrash et al., 2013*; *Sousounis and Tsonis, 2012*). Based on these observations, the newt #19 lenses show a robust transcriptional program as they undergo multiple events of regeneration throughout their lives. Conceptually the process of transdifferentiation might provide robustness to the process of regeneration. In relation to this, it is interesting to note that the transdifferentiation ability of even aged human iris PECs is retained in vitro. Previous studies have shown that such a cell line from an 80-year-old human is capable of transdifferentiating into lens (*Yun et al., 2015*).

In contrast to the patterns observed in the lenses, a comparison of gene expression between the #19 and #0 tails revealed striking differences. Thousands of genes were significantly differentially regulated. Among the most highly deregulated genes in the #19 tail samples were those encoding proteins that are part of the electron transport chain (*Supplementary file 5*). Down-regulation of these gene-sets are part of an established ageing signature in other vertebrates (*López-Otín et al., 2013*; *Zahn et al., 2006*). In addition, several ageing- and senescence-related genes were found to be deregulated in these samples. These findings indicate that the tails of the #19 newts show clear hallmarks of ageing. These observations provide in our opinion strong evidence that a robust transcriptional program ensues after an insult to guarantee that the regenerative ability in newts will not be thwarted with age.

Although the differences in gene expression between the #19 and #0 tails and lenses were pronounced, the irises showed an intermediate pattern, especially with respect to the number of deregulated genes. Similarly to the #19 tails, #19 iris samples down-regulated electron transport chain genes, a sign of ageing. Iris is the source of lens regeneration and ageing of this tissue may hinder the process. Since #19 lenses are comparable to #0 and that is not the case for the iris, there should be a mechanism that amends or corrects the profile from the source (dorsal iris) to the regenerated tissue (lens). *Supplementary file 9* lists genes shown to be differentially regulated in the iris samples and their potential transcriptomic correction in the lens samples. All genes that were found to differ in the iris, the source of lens regeneration, were similar in the regenerate, the lens (*Figure 5* and *Supplementary files 10, 11*). For instance, contig c1229960_g1 corresponding to nicotinamide adenine dinucleotide (NADH)-ubiquinone oxidoreductase chain 1, shows a low

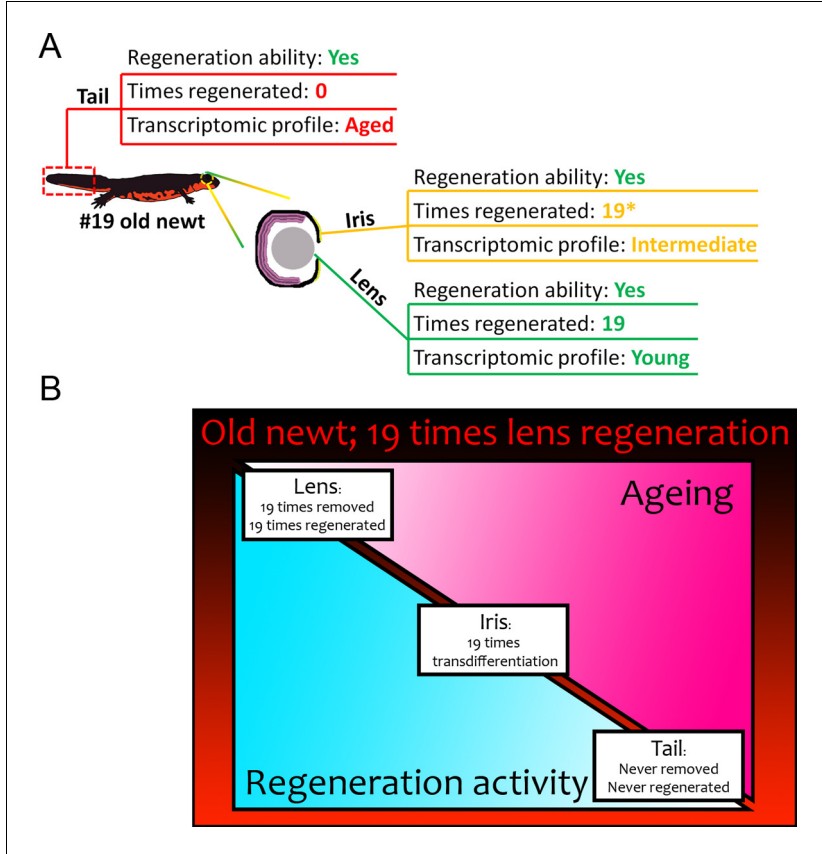

**Figure 7.** Summary of results from our transcriptomic comparisons between #19 and #0 newts. (**A**) Tail samples that had never experienced regeneration showed a marked deregulation of electron transport chain, mitochondrion, and ribosome genes, in #19 newts all signatures of ageing. On the contrary, lenses that were regenerated 19 times over a period of 18 years, showed a transcriptomic profile comparable to never-regenerated lenses from young newts. Iris showed an intermediate profile marked by deregulation of electron transport chain-related genes. (**B**) Regeneration versus ageing in newts. Triangles indicate the amount of regeneration activity (in light blue, decreasing from left to right) and ageing signatures as found by our transcriptomic analysis (in hot pink, increasing from left to right) of sampled #19 tissues. Regeneration initiates a robust transcriptomic program that can be faithfully restarted during repeated insult with no transcriptomic deregulation or molecular signatures of ageing. In our #19 newts, lenses had been fully removed and regenerated 19 times, thus having the highest regeneration activity and showed no signs of ageing. Iris, as the source of lens regeneration, has been regenerated/replenished after transdifferentiation to lens, thus showing some activity and an intermediate profile (the asterisk indicates that iris is not regenerated completely). Tails were never removed or regenerated and showed the most deregulated genes and signatures of ageing compared to the young controls.

expression in the control iris samples, but it is highly expressed in the experimental iris samples. However, the same newts with iris tissues that showed this expression profile had regenerated lenses with low expression of the gene, similar to that of the control iris and lens rendering the difference non-significant. The gene expression differences observed in the iris may also be attributed to the fact that not all dorsal iris PECs contribute to the regeneration of the lost lens. After lentectomy regeneration occurs via dedifferentiation of the lower dorsal tip of the iris. These dorsal iris PECs are either replenished, or cells migrate there from other locations in the iris. Regardless, given that repeated lentectomies always trigger lens regeneration, it is clear that not the whole dorsal iris is eventually transformed into lens cells and that cell proliferation continuously provides the dorsal iris with PECs. Consequently, some parts of the dorsal iris are regenerated and might employ a transcriptomic program similar to that of young controls. On the other hand, other cells might not have this ability and eventually age, thus reflecting the intermediate ageing profile of this tissue. Nevertheless, the cellular or transcriptomic correction of the ageing profile observed in iris to the

regenerated lens should be critical for the integrity of the newly formed organ (*Figure 7*). Recently it has been shown that there is significant turnover of senescent cells during newt limb regeneration. This might explain why newts can regenerate repeatedly their lost structures throughout their lives (*Yun et al., 2015*), As such, the possibility exists that senescent cells are removed from the dorsal iris to ensure the correct process of lens regeneration.

In this study we have compared the same tissues derived from young and old animals. Thus, the differences in the expression profiles were not attributed to the histological complexity. As also discussed above, to compare the ageing status of our collected tissues, our analysis included genes (such as the ones involved in electron transport chain) that are known to be expressed in the majority of cell types and deregulated during ageing. The molecular pathways related to ageing have been studied extensively in other animal models, particularly worms, flies and mice. The use of databases like AGEMAP (a gene expression database for ageing in mice) to make comparisons among species has proven informative for the field of ageing. Many of the genes that are regulated during ageing have been associated with the mitochondrial electron transport chain (*Zahn et al., 2007*; *Signer and Morrison, 2013*; *Gomes et al., 2013*). Another major regulatory pathway involves insulin signaling, which negatively regulates the FOXO transcription factor DAF-16. This transcription factor regulates metabolism and oxidative stress by promoting antioxidant enzymes. The up-regulation of DAF-16 could enhance longevity (*Curran et al., 2009*; *Curran and Ruvkun, 2007*; *Murphy et al., 2003*). Our results suggest that the patterns of ageing in newts are similar to those of other species, particularly those related to the mitochondrial electron transport chain. Thus, it is conceivable that these mechanisms might also be involved in regeneration in newts. Consequently, our 18-year-long experiments provide data that render the newt an indispensable model for addressing issues of regeneration and ageing.

## Materials and methods

### Animal care, handling and surgery

All procedures were performed as described previously (*Sousounis et al., 2014*). *C. pyrrhogaster* was used in this study: Five 32-year-old newts whose lenses had been removed 19 times over a period of 18 years, and five 14-year-old newts that had their original lenses. Tissues collected from every newt were lenses (n = 5), dorsal irises (n = 5), and tails (n = 5). Each tissue from every newt was appropriately labeled and placed in collection tubes.

### RNA extraction

Tissues were stored in RNAlater solution (Ambion, Chicago, Illinois, USA) until RNA isolation. RNA was extracted using an RNeasy Plus Kit (Qiagen, Valencia, CA, USA) according to the manufacturer's protocol.

### Library preparation, RNA sequencing, de novo assembly, and differential gene expression

The input RNA was quantified with a Qubit fluorometric RNA HS assay (Life Technologies, Grand Island, NY, USA). The samples were then analyzed on an Agilent Bioanalyzer using an RNA Pico assay to evaluate the quality. A total of 20 ng of each sample was used to synthesize cDNA using NuGEN Ovation RNA-Seq v2 kit. Libraries were made from 100 ng of cDNA using the NuGEN Ovation Ultralow Library System and then quantified with the Qubit fluorometric DNA HS assay and the Bioanalyzer DNA HS assay. KAPA qPCR was performed to quantify and pool the libraries for sequencing. The libraries were sequenced on a HiSeq 1500 using a 2 x 75 bp high output run. Raw sequencing reads has been deposited in the NCBI's Sequence Read Archive (SRA) database (BioProject accession: PRJNA288378).

Due to the unusually large size of the samples and limited computational resources, Trinity 20140413p1 was used for this project, and we used multi-step in silico normalization for the sequencing reads (*Grabherr et al., 2011*). As the developer suggested, max_cov was set to 50 (personal communication). The final assembly result gave an N50 of 430, and the average contig length was 409. All assembly work was performed on PSC (Pittsburgh Supercomputing Center) Blacklight

which is an SGI UV 1000cc-NUMA shared-memory system comprising 256 blades. The 16 cores on each blade share 128Gb of local memory.

After assembly, the original reads (not in silico normalized) were aligned to the Trinity transcripts to obtain abundance estimates using Bowtie 2 (*Langmead and Salzberg, 2012*). Then, RSEM software was used to estimate the expression levels based on the resulting alignments. After estimating abundance, we obtained the expression profiles for each sample, and the edgeR Bioconductor package was used to identify differentially expressed transcripts (*Robinson et al., 2010*). edgeR analysis was carried out using the protocol of identifying differentially expressed features with biological replicates and counts matrix as abundance estimation pulled from RSEM as input. For the data appearing in *Figures 2A and 4A* differentially expressed genes (p-value<=0.001 and $\log_2$(FC)>=2 in at least one comparison pair) were used. Euclidean distance and complete linkage were used to calculate the correlation. Two subsampling methods were also applied. Jackknife resampling was first used to estimate the variance of the correlation between each pair of samples by systematically leaving out one contig expression from the expression results matrix. We also random selected 20% of contigs to make a subsample, then calculated the correlation matrix of samples. The plots are shown in the *Supplementary files 7 and 8.*

## Annotation, analysis, GO enrichment, and protein network

The de novo assembled transcriptome was annotated against the human reference proteome (e-value<1E-10) using NCBI BLASTx (*Altschul et al., 1990*; *Looso et al., 2013*). The annotated transcripts created the newt reference proteome from which all gene names were derived. Differentially regulated transcripts (FDR<0.05 and FC >2) were mined from the raw edgeR output files and linked to the assigned annotation using custom Perl scripts. For all the analysis using annotated transcripts we used all potential isoform annotations in the testing and reference datasets. For data appearing in *Figure 2B* we selected annotated transcripts expressed more than 1000 average FPKM. The Venn diagram was made with Venny (http://bioinfogp.cnb.csic.es/tools/venny/index.html) and modified with Photoshop (Adobe). For data in *Figure 2C* we used transcripts expressed more than 100 average FPKM in the tissue of interest and less than 100 average FPKM in the other tissues, while the fold change between them was more than 100. For GO enrichment, the UniProt IDs of the differentially regulated gene groups were used as 'gene lists' in the DAVID 6.7 online functional annotation tool (*Huang et al., 2008*, *2009*). We used the newt reference proteome as the source of background genes. We performed the enrichment analysis using the three default gene ontology categories. GO terms with FDR<0.05 were considered enriched. To mine genes related to ageing and/or senescence we searched for gene names with GO terms that contain 'age,, 'aging', 'ageing' or 'senescence' and crossed them with our gene-sets. For the transcriptomic complexity graphs in *Figure 3E*, we sorted average FPKM values from all annotated transcripts individually for each tissue. The percent contribution to the total transcriptomic output was computed by dividing the average FPKM of a certain transcript to the sum FPKM of all transcripts in that tissue. Then transcripts were plotted from the least to the most expressed in a cumulative way (*Supplementary file 1*). To investigate potential signs of ageing in the EL4 sample we performed the following: EL4 genes were considered that deviate from the other #19 samples with the following formula:

$$EL + 2 * \sigma(EL) < EL4 < EL - 2 * \sigma(EL)$$

where EL is the FPKM value of EL1, EL2, EL3, and EL5. Generally, genes were considered that were expressed in the samples if their FPKM value was more than 2 (*Supplementary file 9*). Linear regression analysis appearing in *Figure 5A* was performed with SigmaPlot 11.0 and Excel.

## Acknowledgements

This work was supported by NIH grant EY10540 to PAT. CC was supported by KAKENHI 221S0002 and 24240062.

# Additional information

## Funding

| Funder | Grant reference number | Author |
|---|---|---|
| National Eye Institute | EY10540 | Panagiotis A Tsonis |
| KAKENHI | 221S0002 | Chikafumi Chiba |
| KAKENHI | 24240062 | Chikafumi Chiba |

The funders had no role in study design, data collection and interpretation, or the decision to submit the work for publication.

## Author contributions

KS, Acquisition of data, Analysis and interpretation of data, Drafting or revising the article; FQ, MCY, JLM, FT, CC, Acquisition of data, Analysis and interpretation of data; YE, Contributed reagents; GE, Conception, Acquisition of data, Contributed reagents; PAT, Conception and design, Analysis and interpretation of data, Drafting or revising the article

## Ethics

Animal experimentation: Usage of animals complied with the University of Dayton institutional regulations, IACUC protocol ID 011-02. All surgical procedures were performed under anesthesia with 0.1% ethyl 3-aminobenzoate.

# Additional files

## Supplementary files

• Supplementary file 1. FPKM values of all annotated transcripts and analysis of tissue expression.

• Supplementary file 2. Differential gene expression between lens tissue from repeatedly regenerated lens samples from aged newts and young newts.

• Supplementary file 3. Differential gene expression between aged and young iris samples.

• Supplementary file 4. Differential gene expression between aged and young tail samples.

• Supplementary file 5. Annotation and gene ontology analysis of differentially expressed genes in the tail samples.

• Supplementary file 6. Correlation values between all samples.

• Supplementary file 7. Correlation plot using jackknifing method.

• Supplementary file 8. Correlation plot using 20% random sampling method.

• Supplementary file 9. Analysis of EL4 gene expression to determine signs of ageing.

• Supplementary file 10. Potential correction of the iris profile upon transdifferentiation to lens.

• Supplementary file 11. Annotation and gene ontology analysis of differentially expressed genes in the iris samples.

## Major datasets

The following datasets were generated:

| Author(s) | Year | Dataset title | Dataset ID and/or URL | Database, license, and accessibility information |
| --- | --- | --- | --- | --- |
| Panagiotis A Tsonis | 2015 | Cynops pyrrhogaster ageing transcriptome | http://www.ncbi.nlm.nih.gov/bioproject/288378 | Publicly available at the NCBI Sequence Read Archive (Accession no: PRJNA288378) |

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
