## [Decision Letter]

Thank you for submitting your work entitled "Regeneration thwarts ageing in newts" for peer review at *eLife*. Your submission has been favorably evaluated by Fiona Watt (Senior editor) and four reviewers, one of whom, Alejandro Sánchez Alvarado, is a member of our Board of Reviewing Editors.

The reviewers have discussed the reviews with one another and the Reviewing editor has drafted this decision to help you prepare a revised submission.

Summary:

The present manuscript by Sousounis et al., reports on a unique lens regeneration longitudinal study (⩾18 years) aimed at determining whether age may or may not affect regenerative capacities. The central merit of this work is that it provides experimental data to challenge/confirm previously held assumptions, an essential aspect needed for the growth and maturation of the field of regeneration.

In essence, the authors' data indicate that after 19 annual lens regeneration events, the iris transcriptional profile remains statistically unchanged when compared to a younger RNA sample (14 vs. 14⩽32 year old newts). In addition, the authors also examined tail regeneration samples, in which they did detect transcriptional changes associated with age. These findings are significant to our understanding of lens regeneration. The identification from the gene expression data of an intermediate pattern of transcription indicates that a specialized subpopulation of PECs in the dorsal iris contribute to lens regeneration and that the process of regeneration itself may result in the generation of a metastable state for the contributing cells which then perpetuates their regenerative capacities. If anything, this work definitively and unambiguously demonstrate the principle of tissue identity in the process of Wolffian regeneration, i.e., transdifferentiation.

Essential revisions:

There are, however, a few issues that the authors should be able to address in order to strengthen the current manuscript. They are as follows:

1) Given that the youngest sample is from a 14 year old animal, and that no lentectomized 32 year old lens transcriptome is presented, what is the evidence for "rejuvenation"? The term is distracting and inaccurate for the data being presented. No data is presented supporting/demonstrating that unamputated newt lenses age. The reference to Sousounis and Tsonis (2012) used in the article to support this claim is about human lenses, not amphibians. The results suggest, instead, that the transcriptional program of the lens is more robust/canalized against change over time (aging), than those of the iris and tail. Because the emphasis on aging is an extension of the data, whereas the inference about multiple episodes through the lifespan is directly established, the reviewers agree that the aging be de-emphasized, starting with changing the title of the article to one which is closer to the experimental findings.

2) Given the vastly different nature of the tissues being sampled (iris, lens and tail), it seems appropriate to request from the authors additional discussion regarding the possibility that the histological complexity of the samples being compared and the attendant methods of data analyses may or may not have contributed to the observed results.

3) The manuscript is very concise, so there is certainly room for additional material. It is disappointing that it ends without any mechanistic analysis or hypothesis about their results with the lens, e.g., is it something to do with the origin of the regenerated lens by transdifferentiation, and are there any suggestions from experimental transdifferentiation in mammalian cells that would help with this? It presumably is not due to salamanders having young blood, or removing senescent cells, so what might be happening?

4) In Figure 3, it appears that sample EL4 seems to exhibit a different pattern, with changes that seem to be comparable to samples that show variation with time such as tails and irises. Does this particular lens show signs of ageing? Please comment on this. Some discussion of clusters of similarly expressed genes would be informative. For example, at the bottom of the heat map in Figure 2 is a cluster of genes that are uniquely, highly expressed in lens – what are these genes? Much more could be said about the data than "a nearly uniform pattern." Also, formally (I don't think it will change the results in this case) one should perform some type of resampling method (e.g. jackknifing) of the dendrogram to show statistical support for the sample groupings.

5) In the iris samples the authors note that there are 30 up-regulated and 24 down-regulated genes comparing the old and young samples. It would be helpful to state if any of these are expressed in the lens, and if so, whether they are indistinguishable in expression between the 0 and 19 lenses. This seems an important point as it would suggest some kind of correction, at whatever level, during the process of regeneration from the iris.

6) There is a lack of validation of the transcriptomic analysis. This arises particularly with the tail and iris samples. It would be helpful to present RT-PCR data for a few key up and down regulated genes.

---

## [Author Response]

*Essential revisions: There are, however, a few issues that the authors should be able to address in order to strengthen the current manuscript. They are as follows:*

*1) Given that the youngest sample is from a 14 year old animal, and that no lentectomized 32 year old lens transcriptome is presented, what is the evidence for "rejuvenation"? The term is distracting and inaccurate for the data being presented. No data is presented supporting/demonstrating that unamputated newt lenses age. The reference to Sousounis and Tsonis (2012) used in the article to support this claim is about human lenses, not amphibians. The results suggest, instead, that the transcriptional program of the lens is more robust/canalized against change over time (aging), than those of the iris and tail. Because the emphasis on aging is an extension of the data, whereas the inference about multiple episodes through the lifespan is directly established, the reviewers agree that the aging be de-emphasized, starting with changing the title of the article to one which is closer to the experimental findings.*

The reviewers raise an important issue and we agree that our main results do show a robust program in the repeatedly regenerated lens and that we cannot claim “rejuvenation”. So, as suggested, we have de-emphasized the ageing aspect throughout the manuscript. We have changed the title as suggested and we have accordingly edited the manuscript to reflect this. However, we have to mention that our transcriptomic analysis does compare old with young newts as well regenerated and non-regenerated tissues of old newts and thus some discussion of ageing should come into place as well. We have also emphasized the fact that there is a lot of literature showing the detrimental effects of ageing in the vertebrate lens in an attempt to strengthen our conclusions about ageing and regeneration in our model.

*2) Given the vastly different nature of the tissues being sampled (iris, lens and tail), it seems appropriate to request from the authors additional discussion regarding the possibility that the histological complexity of the samples being compared and the attendant methods of data analyses may or may not have contributed to the observed results.*

This is an interesting question, however, we think that this is not the case. The differences in expression profiles should not be attributed to the histological complexity between tail, iris and lens because we compare tail with tail, iris with iris and lens with lens. The only variable was that one set was from young animals and the other from old (with or without regeneration). We have added discussion on this issue.

*3) The manuscript is very concise, so there is certainly room for additional material. It is disappointing that it ends without any mechanistic analysis or hypothesis about their results with the lens, e.g., is it something to do with the origin of the regenerated lens by transdifferentiation, and are there any suggestions from experimental transdifferentiation in mammalian cells that would help with this? It presumably is not due to salamanders having young blood, or removing senescent cells, so what might be happening?*

Since, following the reviewers’ advise, we have de-emphasized the ageing issue and because of additional analysis requested for comments 4 and 5 (see below) the paper has been considerably expanded. Also some other graphs have been added to better show the robust transcriptional program in the lens vs. iris and tail. Additional panels are presented in Figure 2, Figure 3 and Figure 4. The Discussion section of the manuscript has been increased and we added Figure 7 that illustrates the main issues and conclusions raised in our study. The suggestion about the origin of regenerated lens by transdifferentiation is interesting. We have also added this to the Discussion and as the reviewers pointed in relation to experiments with human PECs.

*4) In Figure 3, it appears that sample EL4 seems to exhibit a different pattern, with changes that seem to be comparable to samples that show variation with time such as tails and irises. Does this particular lens show signs of ageing? Please comment on this. Some discussion of clusters of similarly expressed genes would be informative. For example, at the bottom of the heat map in Figure 2 is a cluster of genes that are uniquely, highly expressed in lens – what are these genes? Much more could be said about the data than "a nearly uniform pattern." Also, formally (I don't think it will change the results in this case) one should perform some type of resampling method (e.g. jackknifing) of the dendrogram to show statistical support for the sample groupings.*

In Figure 2 we have added two more panels identifying the genes that are highly expressed or regulated in each tissue. We have added also a new Supplementary file ([Supplementary-material SD1-data] in the revised version).

With regard the EL4 sample: the reviewers are right – EL4 is somewhat different than the others. True enough after looking at the correlation matrix between all samples ([Supplementary-material SD6-data]), we find sample EL4 to have the lowest value among the #19 lenses, something that was expected by observing the plot. However, it was higher than the correlation between #19 irises or tails with the #0 equivalent tissues ([Supplementary-material SD6-data]). That is an indication that sample EL4 is not at the same level as the iris and tail samples. We did an analysis with crystallin-related genes. Crystallin-related genes were extracted based on the annotation of the individual transcripts and then gene expression was analyzed in the #19 lens samples. We found around 30% of the genes to be deregulated in EL4 sample versus the other #19 lens samples. However, the crystallin genes were shown to be up-regulated in the EL4 sample, something that does not correlate to the regulation of crystallin genes during disease, like cataract, that crystallin genes are usually down-regulated. Overall, our analysis of the potential ageing profile of the EL4 sample shows that yes indeed, there is a deviation of gene expression but this does not seem to be due to the ageing of this particular sample. This analysis is mentioned now in the manuscript presented in [Supplementary-material SD9-data].

We have performed jackknifing and 20% re-sampling analysis and we find no difference in the data. We show this analysis in [Supplementary-material SD7-data] and [Supplementary-material SD8-data].

*5) In the iris samples the authors note that there are 30 up-regulated and 24 down-regulated genes comparing the old and young samples. It would be helpful to state if any of these are expressed in the lens, and if so, whether they are indistinguishable in expression between the 0 and 19 lenses. This seems an important point as it would suggest some kind of correction, at whatever level, during the process of regeneration from the iris.*

This is a very interesting comment. Since between #0 and #19 lenses there are not any differentially regulated genes, all the genes that are shown to be differentially regulated in the iris are in fact “corrected” in the regenerated lens. This is an important point for which we have added a new figure (Figure 5) with appropriate text and [Supplementary-material SD10-data]. Indeed we show that the values found to be differentially expressed between iris samples are corrected to normal in the lens (we use all 311 regulated genes).

*6) There is a lack of validation of the transcriptomic analysis. This arises particularly with the tail and iris samples. It would be helpful to present RT-PCR data for a few key up and down regulated genes.*

Unfortunately we do not have any of the samples. Due to the small size and numbers (2 lenses per sample) all the collected tissues were used for RNA sequencing. But based on the robustness we see in all 5 samples from each group we are confident that the data are valid.